# The predatory soil bacterium *Lysobacter* reprograms quorum sensing system to regulate antifungal antibiotic production in a cyclic-di-GMP-independent manner

Kaihuai Li [1,2], Gaoge Xu[1], Bo Wang[1], Guichun Wu[1], Rongxian Hou[1,2] & Fengquan Liu [1,2✉]

Soil bacteria often harbour various toxins to against eukaryotic or prokaryotic. Diffusible signal factors (DSFs) represent a unique group of quorum sensing (QS) chemicals that modulate interspecies competition in bacteria that do not produce antibiotic-like molecules. However, the molecular mechanism by which DSF-mediated QS systems regulate antibiotic production for interspecies competition remains largely unknown in soil biocontrol bacteria. In this study, we find that the necessary QS system component protein RpfG from *Lysobacter*, in addition to being a cyclic dimeric GMP (c-di-GMP) phosphodiesterase (PDE), regulates the biosynthesis of an antifungal factor (heat-stable antifungal factor, HSAF), which does not appear to depend on the enzymatic activity. Interestingly, we show that RpfG interacts with three hybrid two-component system (HyTCS) proteins, HtsH1, HtsH2, and HtsH3, to regulate HSAF production in *Lysobacter*. In vitro studies show that each of these proteins interacted with RpfG, which reduced the PDE activity of RpfG. Finally, we show that the cytoplasmic proportions of these proteins depended on their phosphorylation activity and binding to the promoter controlling the genes implicated in HSAF synthesis. These findings reveal a previously uncharacterized mechanism of DSF signalling in antibiotic production in soil bacteria.

[1] Institute of Plant Protection, Jiangsu Academy of Agricultural Sciences, Jiangsu Key Laboratory for Food Quality and Safety-State Key Laboratory Cultivation Base, Ministry of Science and Technology, 210014 Nanjing, China. [2] College of Plant Protection, Nanjing Agricultural University, 210095 Nanjing, China. ✉email: fqliu20011@sina.com

Interspecies competition plays a key role in shaping microbial populations and determining the bacterial species that are dominant in a given niche[1]. To fend off these competitors, bacteria deploy various toxins against eukaryotic or prokaryotic. The diffusible signal factor (DSF) family signals are important for maintaining the interspecies competitive fitness of bacteria[2]. The DSF family signals interfere with the morphological transition of *Candida albicans* through inter-kingdom communication[3,4].

DSFs represent a class of widely conserved QS signals with a fatty acid moiety that regulate various biological functions in pathogenic and beneficial environmental bacteria[5,6]. The Rpf gene cluster is important for the DSF signaling network in bacteria, and the role of the RpfF and RpfC/RpfG two-component system (TCS) in this gene cluster in DSF production and signal transduction has been well documented[5,7–10]. Several lines of evidence indicate that RpfC and RpfG constitute a TCS responsible for the detection and transduction of the QS signal DSF[5,11]. RpfC undergoes autophosphorylation upon sensing high levels of extracellular DSF signals[5,9,11]. A previous study revealed that RpfG contains both an N-terminal response regulator domain and a C-terminal HD-GYP domain[12]. The activated HD-GYP domain of RpfG has cyclic dimeric GMP (c-di-GMP) phosphodiesterase (PDE) activity that can degrade c-di-GMP, an inhibitory ligand of the global transcription factor Clp. Consequently, derepressed Clp drives the expression of several hundred genes, including those encoding virulence factor production in the plant pathogen *Xanthomonas*[11,13–15]. The DSF signal family is a structural class of QS signals with the cis-2-unsaturated fatty acid moiety. Surprisingly, a DSF-like signal (LeDSF3), unlike other members of the DSF family, does not contain the cis double bond, and has been characterized as a QS signal in the biocontrol agent strain *Lysobacter enzymogenes*[16]. However, the regulatory mechanism of the DSF-mediated QS system remains unknown in bacteria that are beneficial to plants.

*L. enzymogenes* is a nonpathogenic strain that was used to control crop fungal diseases known for the synthesis of an antifungal factor (heat-stable antifungal factor, HSAF) that exhibits inhibitory activity against a wide range of fungal species[17–24]. Our previous work revealed that RpfG affects production of the antifungal factor HSAF in *L. enzymogenes*[25]. However, the molecular mechanism by which RpfG regulates the biosynthesis of HSAF remains unknown.

In the present study, we found that unlike the *Xanthomonas* RpfC/RpfG-Clp signaling pathway, the *L. enzymogenes* RpfG protein interacts with three hybrid two-component system (HyTCS) proteins (HtsH1, HtsH2, and HtsH3) to regulate the production of the antifungal factor HSAF and describe their regulatory functions in soil bacteria. The HtsH1, HtsH2, and HtsH3 functions likely represent a common mechanism that helps establish signaling specificity in bacteria for interspecies competition.

## Results

### The HD-GYP domain of RpfG has PDE activity and can degrade c-di-GMP.
Sequence analysis revealed that the HD-GYP domain contains all residues essential for PDE activities, thus suggesting that RpfG may be a PDE enzyme. HD-GYP domain-containing proteins can degrade the c-di-GMP to GMP and 5′-pGpG. However, the in vitro enzyme activity of RpfG homologs has not been studied and identified. To obtain direct evidence for the biochemical function of RpfG, recombinant N-terminal maltose binding protein (MBP) RpfG (designated RpfG-MBP) was produced. The proteins had a monomeric molecular weight of 71 kDa, as observed by SDS gel electrophoresis, and were purified by Dextrin Sepharose High Performance to obtain the

preparations (Fig. 1a and Supplementary Fig. 10). The RpfG protein was fused with the MBP tag, leading to the presence of some impurities. This RpfG-MBP protein was able to degrade the model substrate c-di-GMP to 5′-pGpG, consistent with its PDE activity (Fig. 1b). Quantitative analysis revealed that RpfG-MBP exhibited a high level of activity for the degradation of c-di-GMP with 100% degraded at 5 min after initiation of the reaction in comparison to the MBP enzyme as a control (Fig. 1c). To better understand the roles of the HD-GYP domain in RpfG function, we substituted the RpfG residues His-190, Asp-191, Gly-253, Tyr-254, and Pro-255 of the HD-GYP signature motif with alanine (Ala) by site-directed mutagenesis into constructs expressing the RpfG-H190A-MBP, RpfG-D191A-MBP, RpfG-G253A-MBP, and RpfG-P255A-MBP proteins. We tested the c-di-GMP PDE activity of these mutant proteins. Our results showed that point mutations of the His-190, Asp-191, Gly-253, and Pro-255 residues in RpfG almost abolished PDE activity (Fig. 1b, c). These data suggest that RpfG has PDE activity and that the HD-GYP individual residues are required for full PDE activity of RpfG in vivo.

### HSAF production does not depend on RpfG PDE enzymatic activity.
Previous studies found that the QS signal LeDSF3 positively regulates HSAF biosynthesis[16]. To ultimately determine whether DSF type-based QS systems are critical for regulating the synthesis of HSAF, we quantified the HSAF production in the Δ*rpfF* and Δ*rpfG* mutant stains grown in 10% TSB medium or 10% TSB medium supplemented with 10 μM canonical DSF. As shown in Supplementary Fig. 1, HSAF production in the Δ*rpfF* and Δ*rpfG* mutant strains was completely suppressed, and DSF significantly rescued HSAF production in the Δ*rpfF* mutant strain, but did not rescued that by the Δ*rpfG* mutant strain. These findings suggested that DSF type-based QS systems are critical for regulating the biosynthesis of HSAF in *L. enzymogenes*. Our previous work also revealed that RpfG affects HSAF production in *L. enzymogenes*[25]. However, the molecular mechanism by which RpfG regulates HSAF synthesis remains unknown. Since RpfG, as a PDE enzyme, was able to degrade the substrate c-di-GMP to 5′-pGpG, we investigated whether RpfG PDE activity played a major role on controlling the production of HSAF in *L. enzymogenes*. We quantified HSAF production in the Δ*rpfG* mutant and complementary strain (Δ*rpfG/rpfG*) by HPLC. We found that HSAF production in the Δ*rpfG* mutant strain was completely suppressed (Fig. 2a). The complementary strain Δ*rpfG/rpfG* yielded HSAF at the level of the wild-type strain (Fig. 2a). These results were similar to those of the above research[25]. To examine the relationship between the regulatory and enzymatic activities of RpfG, mutations at the conserved His-190, Asp-191, Gly-253, Tyr-254, and Pro-255 of the HD-GYP signature motif with alanine (Ala) were examined by site-directed mutagenesis. We tested HSAF production in the Δ*rpfG* mutant strain carrying plasmids encoding these mutant proteins. The strains expressing the RpfG H190A, D191A, G253A, Y254A, and P255A mutant proteins showed increased HSAF production compared with the Δ*rpfG* mutant strain. These results were superior to those of the complementary strain Δ*rpfG/rpfG*. Importantly, the Δ*rpfG* mutant strain and complemented strains (Δ*rpfG/rpfG*, Δ*rpfG/rpfG* H190A, Δ*rpfG/rpfG* D191A, Δ*rpfG/rpfG* G253A, Δ*rpfG/rpfG* Y254A, and Δ*rpfG/rpfG* P255A) did not impair bacterial growth (Fig. 2b, c). As described above, His-190, Asp-191, Gly-253, Tyr-254, and Pro-255 were found to be critical for the PDE activity of RpfG (Fig. 1), implying that HSAF is regulated in a PDE independent manner. To test this prediction, we compared intracellular c-di-GMP concentrations in the Δ*rpfG* mutant and the wild type and in the HSAF-production medium

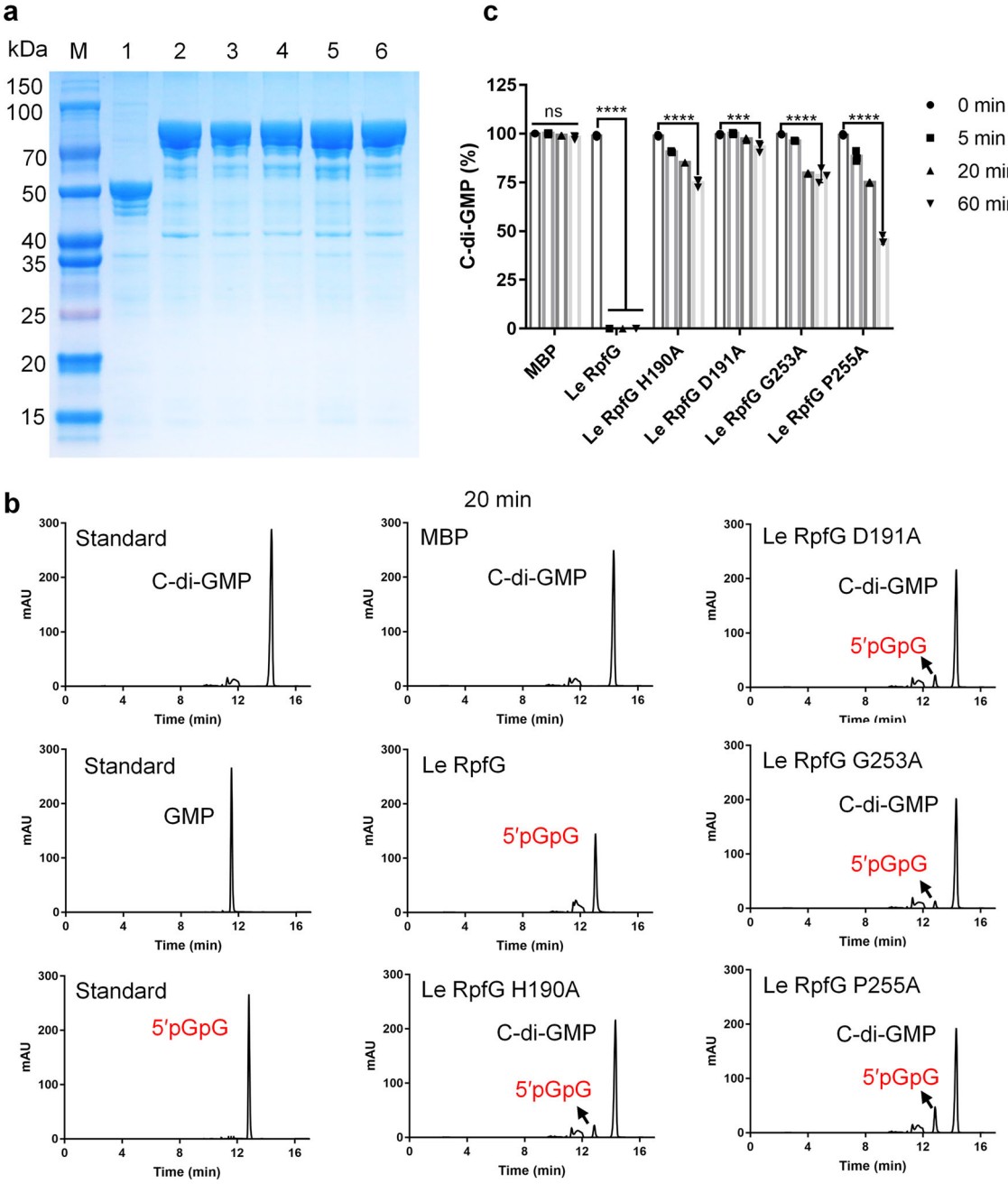

**Fig. 1 RpfG is a PDE enzyme. a** The purified protein was analysed by 12% SDS-PAGE. Lane M, molecular mass markers; lane 1, MBP protein; lane 2, RpfG-MBP protein; lane 3, RpfG-MBP H190A protein; lane 4, RpfG-MBP D191A protein; lane 5, RpfG-MBP G253A protein; lane 6, RpfG-MBP P255A protein. **b** The purified protein has enzymatic activity against c-di-GMP. HPLC analysis of aliquots of reaction mixtures boiled immediately after addition of the enzyme and after 20 min of incubation shows the degradation of c-di-GMP to a compound with the same mobility as the 5′pGpG standard. **c** The purified RpfG protein has PDE enzymatic activity. For convenience of comparison, the peak of c-di-GMP in the RpfG solution at 0 min was defined as 100% and used to normalize the c-di-GMP level ratios at different time points. Error bars, means ± standard deviations ($n = 3$ biologically independent samples). ***$P < 0.001$, ****$P < 0.0001$, assessed by one-way ANOVA. All experiments were repeated three times with similar results.

(10% TSB). We found that the concentration in the Δ*rpfG* mutant did not significantly change c-di-GMP production compared with the wild-type strain (Fig. 2d). These findings indicated that the regulatory activity of RpfG does not depend on its PDE enzymatic activity against c-di-GMPs.

**RpfG binds directly to the HyTCS protein HtsHs.** The above findings confirmed that RpfG does not regulate HSAF biosynthesis through the c-di-GMP signaling pathway, indicating that RpfG might regulate HSAF biosynthesis through interactions

with other proteins in *L. enzymogenes*. To further explore the mechanisms underlying the contribution of RpfG to HSAF production, we used a bioinformatic tool (STRING) to identify potential interactors for RpfG; these represent interactions that possibly lead to alterations in HSAF synthesis. We discovered through bioinformatics predictions that RpfG interacts with three HyTCS proteins (Supplementary Fig. 2a). We designated the HyTCS protein HtsH (hybrid two-component signaling system regulating HSAF production) based on the findings of this study. To verify the operon structure of *htsHs* for in-depth genetic

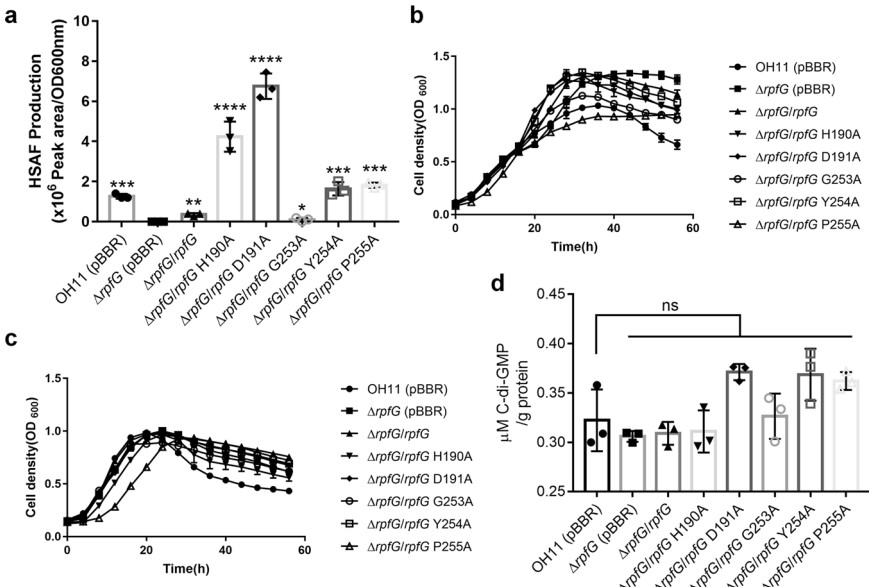

**Fig. 2 RpfG-mediated regulation of HSAF production does not depend on its PDE activity. a** Quantification of HSAF produced by the *rpfG* mutant strain and strains complemented with *rpfG* or the *rpfG* site-directed mutant genes grown in 10% TSB medium. **b** Growth curves of the *rpfG* mutant strain and strains complemented with *rpfG* or the *rpfG* site-directed mutant genes in rich LB medium. **c** Growth curves of the *rpfG* mutant strain and strains complemented with the *rpfG* or *rpfG* site-directed mutant genes in 10% TSB medium. **d** Intracellular c-di-GMP concentrations in the wild type, *rpfG* mutant and strains complemented with *rpfG* or the *rpfG* site-directed mutant genes. Error bars, means ± standard deviations ($n = 3$ biologically independent samples). *$P < 0.05$, **$P < 0.01$, ***$P < 0.001$, ****$P < 0.0001$, assessed by one-way ANOVA. All experiments were repeated three times with similar results.

analyses, a series of RT-PCR primers (Supplementary Table 2 and Supplementary Fig. 2b) were designed to determine whether there are intergenic transcripts crossing the adjacent genes. As shown in Supplementary Fig. 2c, Le3071 (*htsH1*), Le3072 (*htsH2*), and Le3073 (*htsH3*) likely constitute a single transcription unit because the corresponding intergenic transcripts were successfully amplified. Le3071 (*htsH1*), Le3072 (*htsH2*), and Le3073 (*htsH3*) encode a group of typical HyTCS proteins with pfam Reg_prop, pfam Y-Y-Y, HisKA, HATPase_c, and Response_reg domains. All three HyTCS proteins contain one predicted transmembrane region (Supplementary Fig. 2d). We examined the alignments of three HyTCS proteins (HtsH1, HtsH2, and HtsH3), and the results showed that the HtsH1 protein shares 50% and 53% identity with HtsH2 and HtsH3, respectively. We also aligned HtsH2 with HtsH3, and the identity values were 50% (Supplementary Figure 3).

To examine whether RpfG could directly bind to the HyTCS proteins (HtsH1, HtsH2 and HtsH3), we used a pull-down assay using *E. coli*-expressed proteins in vitro. We purified recombinant RpfG-MBP and the cytoplasmic fragments of HtsH1, HtsH2, and HtsH3 (HtsH1C-Flag-His, HtsH2C-HA-His, and HtsH3C-Myc-His, respectively) from *E. coli* (Fig. 1a and Supplementary Figure 4, 10). First, we tested the ability of RpfG-MBP to pull down HtsH1C-Flag-His, HtsH2C-HA-His, and HtsH3C-Myc-His. Finally, we confirmed the interaction between RpfG and HtsH1, HtsH2 or HtsH3 using pull down assays (Fig. 3a and Supplementary Fig. 11). Conversely, we examined the ability of HtsH1C-Flag-His, HtsH2C-HA-His, and HtsH3C-Myc-His to pull down RpfG-MBP and observed a positive signal (Fig. 3b–d and Supplementary Fig. 11). Second, we used surface plasmon resonance (SPR) to measure the possible binding events between the RpfG and HtsH1C, HtsH2C, or HtsH3C proteins. The RpfG-MBP sensor physically bound HtsH1C-Flag-His with a binding constant ($K_D$) of 0.06675 μM (Fig. 3e), suggesting an intermediate level of protein–protein interaction. We also confirmed direct binding between the RpfG-MBP and HtsH2C-HA-His or HtsH3C-Myc-His proteins by SPR ($K_D = 0.2998$ μM or $K_D =$

0.1678 μM, respectively) (Fig. 3f, g). Additionally, the SPR assay revealed that HtsH1C-Flag-His bound to HtsH2C-HA-His or HtsH3C-Myc-His with reasonably high affinity ($K_D = 0.09619$ μM or $K_D = 0.1597$ μM, respectively), and revealed that HtsH2C-HA-His bound HtsH3C-Myc-His with a $K_D$ value of 0.1782 μM (Supplementary Fig. 5). Taken together, these experiments demonstrate that RpfG directly interacts with HtsH1, HtsH2, or HtsH3 proteins in vitro.

**HtsHs inhibits the PDE enzymatic activity of RpfG**. We wondered whether RpfG and HtsH1, HtsH2, or HtsH3 interactions affect the PDE activity of RpfG. To test this hypothesis, we used a biochemical assay in which c-di-GMP hydrolysis by RpfG-MBP was assayed in the absence or presence of HtsH1, HtsH2, or HtsH3 at concentrations ranging from 0 to 32 μM. The results of the assay showed that the PDE activity of RpfG-MBP was lower in the presence than in the absence of HtsH1C-Flag-His, HtsH2C-HA-His, or HtsH3C-Myc-His (Fig. 4). Therefore, the results of the assays suggested that HtsH1, HtsH2, and HtsH3 inhibited the PDE enzymatic activity of RpfG. This result further confirms that the ability of RpfG to regulate HSAF production does not depend on its PDE enzymatic activity against c-di-GMP in *L. enzymogenes*.

**Deletion of *htsH1*, *htsH2*, and *htsH3* resulted in decreased HSAF production**. The above studies indicated that RpfG might regulate HSAF biosynthesis by interacting with HtsHs (HtsH1, HtsH2, and HtsH3) in *L. enzymogenes*. To identify the physiological functions of HtsH1, HtsH2, and HtsH3 in HSAF production, the genes *htsH1* (Le3071), *htsH2* (Le3072), and *htsH3* (Le3073) were deleted using a two-step homologous recombination approach to construct the single knockout strains Δ*htsH1*, Δ*htsH2*, and Δ*htsH3*; the double mutant knockout strains Δ*htsH12*, Δ*htsH13*, and Δ*htsH23*; and the triple knockout strain Δ*htsH123*. Using quantitative RT-PCR (qRT-PCR), we measured the mRNA abundance of *htsH1*, *htsH2*, *htsH3*, Le3074 and

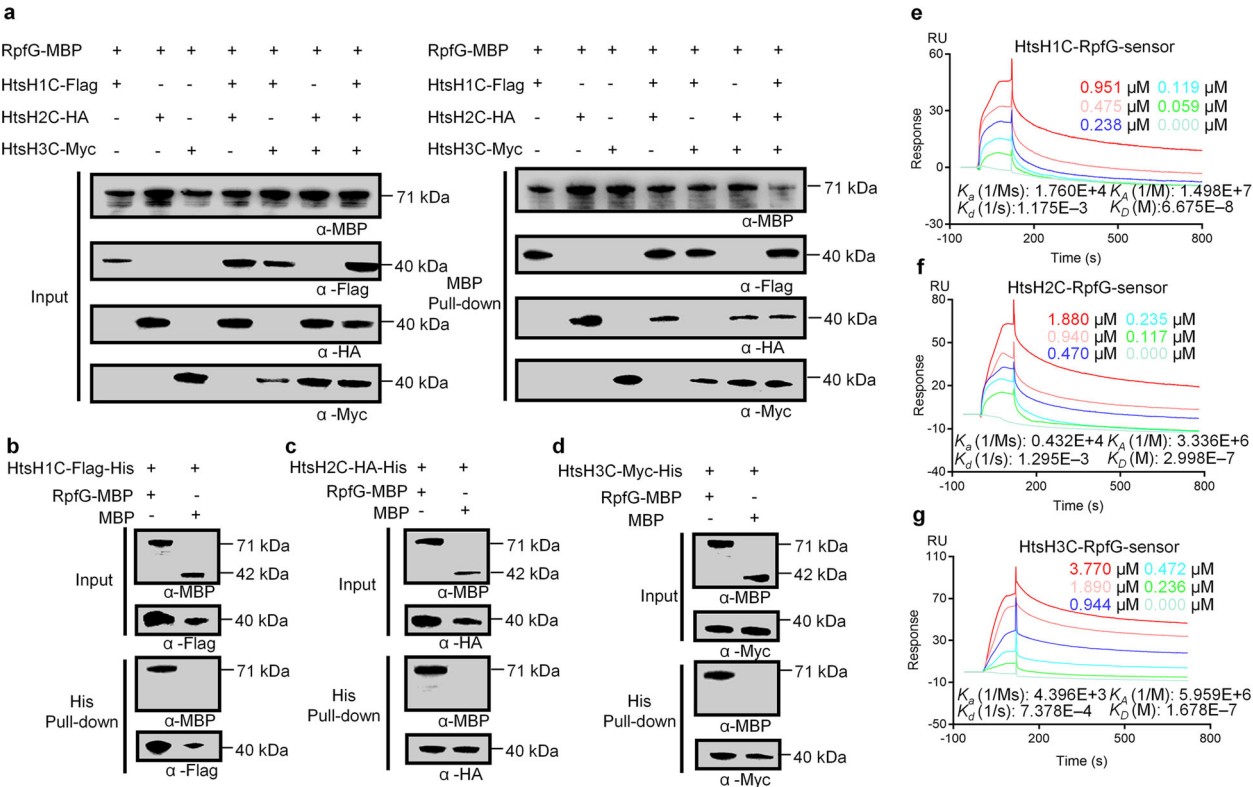

**Fig. 3 RpfG interaction with HtsH1, HtsH2, and HtsH3. a** An MBP pull-down assay confirming interactions between RpfG-MBP and the cytoplasmic fragment of HtsH1, HtsH2, and HtsH3 (HtsH1C-Flag-His, HtsH2C-HA-His and HtsH3C-Myc-His, respectively). The pull-down assay was carried out using anti-MBP antibody. Western blotting was performed using anti-MBP, anti-Flag, anti-HA, and anti-Myc antibodies. **b** A His pull-down assay confirming interactions between the cytoplasmic fragment of HtsH1 (HtsH1C-Flag-His) and RpfG-MBP. The pull-down assay was carried out using Ni-nitrilotriacetic acid (NTA) agarose. Western blotting was performed using anti-Flag and anti-MBP antibodies. **c** A His pull-down assay confirming interactions between the cytoplasmic fragment of HtsH2 (HtsH2C-HA-His) and RpfG-MBP. The pull-down assay was carried out using Ni-NTA agarose. Western blotting was performed using anti-HA and anti-MBP antibodies. **d** A His pull-down assay confirming interactions between the cytoplasmic fragment of HtsH3 (HtsH3C-Myc-His) and RpfG-MBP. The pull-down assay was carried out using Ni-NTA agarose. Western blotting was performed using anti-Myc and anti-MBP antibodies. **e** SPR showing that HtsH1C-Flag-His forms a complex with RpfG-MBP with $K_D = 0.06675\,\mu M$. **f** SPR showing that HtsH2C-HA-His forms a complex with RpfG-MBP with $K_D = 0.2998\,\mu M$. **g** SPR showing that HtsH3C-Myc-His forms a complex with RpfG-MBP with $K_D = 0.1678\,\mu M$.

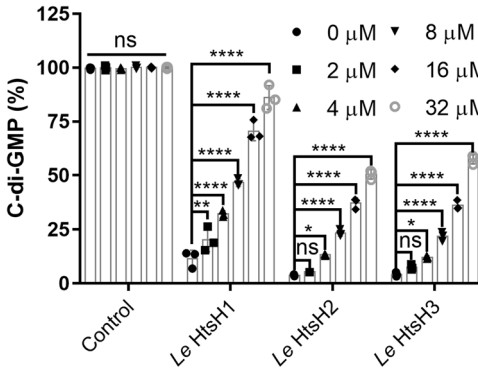

**Fig. 4 Impact of HtsH1, HtsH2, and HtsH3 on RpfG enzyme activity.** The addition of HtsH1C-Flag-His, HtsH2C-HA-His, and HtsH3C-Myc-His proteins to an RpfG-MBP protein solution decreased its c-di-GMP phosphodiesterase activity. For convenience of comparison, the peak of c-di-GMP in the MBP solution was defined as 100% and used to normalize the c-di-GMP level ratios at 20 min.

Le3075 in the *htsHs* mutants (Δ*htsH1*, Δ*htsH2*, Δ*htsH3* and Δ*htsH123*), and found that the genes downstream of the *htsHs* gene deletion mutants were expressed (Supplementary Fig. 6). Subsequently, we quantified HSAF production in all the above mutant strains described by HPLC. We found that Δ*htsH1*,

Δ*htsH2*, and Δ*htsH3* exhibited slightly decreased HSAF levels. However, the double-mutant strains Δ*htsH12*, Δ*htsH13*, Δ*htsH23*, and triple-mutant strain Δ*htsH123* exhibited a significant decrease in HSAF levels compared with the wild-type levels (Fig. 5a). To determine the role of HtsH1, HtsH2, and HtsH3 in the regulation of HSAF biosynthesis, we complemented Δ*htsH1*, Δ*htsH2*, Δ*htsH3*, the double-mutant strains Δ*htsH12*, Δ*htsH13*, Δ*htsH23*, and the triple-mutant strain Δ*htsH123* with plasmid-borne *htsH1*, *htsH2*, *htsH3*, *htsH12*, *htsH13*, *htsH23*, and *htsH123*. HSAF production in the complemented strains (Δ*htsH1*/*htsH1*, Δ*htsH2*/*htsH2*, Δ*htsH3*/*htsH3*, Δ*htsH12*/*htsH12*, Δ*htsH13*/*htsH13*, Δ*htsH23*/*htsH23*, and Δ*htsH123*/*htsH123*) restored HSAF biosynthesis compared with the wild-type levels (Fig. 5a). Importantly, Δ*htsH1*, Δ*htsH2*, and Δ*htsH3*; the double-mutant strains Δ*htsH12*, Δ*htsH13*, and Δ*htsH23*; and the triple-mutant strain Δ*htsH123* did not exhibit impaired bacterial growth (Fig. 5b, c), implying that HtsH1, HtsH2, and HtsH3 play a specific role in regulating HSAF production.

**HtsH1, HtsH2, and HtsH3 positively regulate HSAF biosynthesis gene expression.** Earlier, we found that deletion of *htsH1*, *htsH2*, and *htsH3* resulted in decreased HSAF production. Thus, we wondered whether HtsH1, HtsH2, and HtsH3 might directly target HSAF biosynthesis gene promoters. To test this hypothesis, we performed an *E. coli*-based one-hybrid assay. As

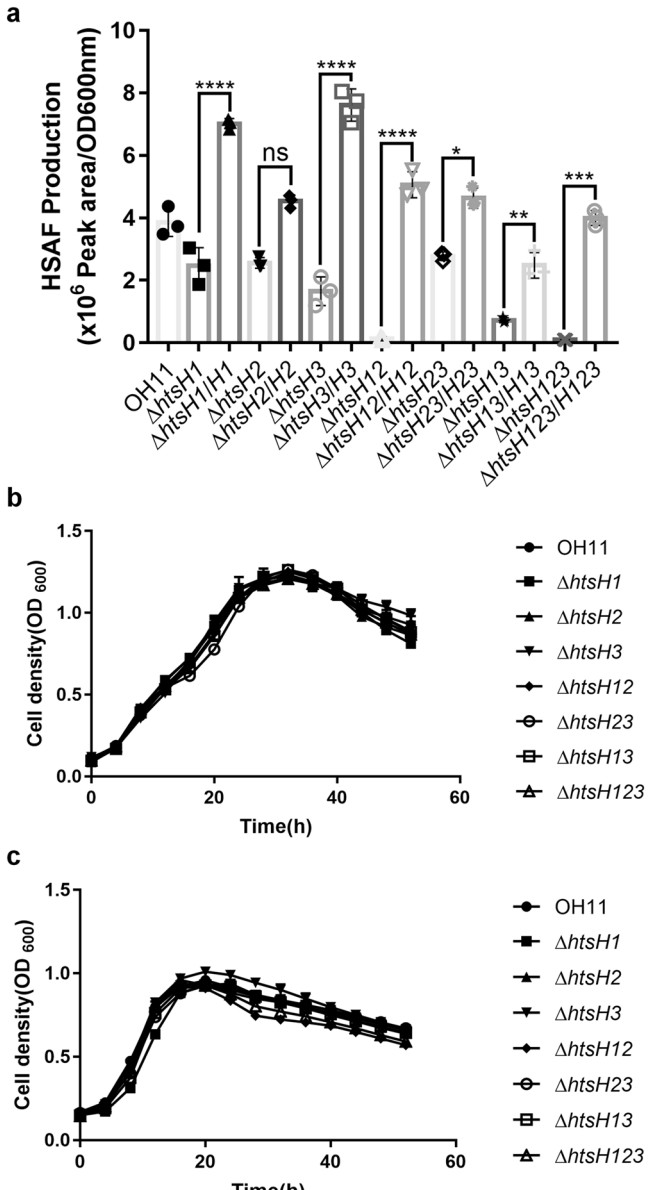

**Fig. 5 Deletion of *htsH1*, *htsH2*, and *htsH3* resulted in decreased HSAF production. a** Quantification of the HSAF produced by the *htsH1*, *htsH2*, and *htsH3* mutant strains and strains complemented with the *htsH1*, *htsH2*, and *htsH3* genes grown in 10% TSB medium. **b** Growth curves of the *htsH1*, *htsH2*, and *htsH3* mutant strains and strains complemented with the *htsH1*, *htsH2*, and *htsH3* genes in LB-rich medium. **c** Growth curves of the *htsH1*, *htsH2*, and *htsH3* mutant strains and strains complemented with the *htsH1*, *htsH2*, and *htsH3* genes in 10% TSB medium. Error bars, means ± standard deviations ($n = 3$ biologically independent samples). *$P < 0.05$, **$P < 0.01$, ***$P < 0.001$, ****$P < 0.0001$, assessed by one-way ANOVA. All experiments were repeated three times with similar results.

shown in Fig. 6a, HtsH1, HtsH2, and HtsH3 directly bound to the upstream region of the HSAF biosynthesis operon (p*lafB*).

Next, we tested the ability of HtsH1, HtsH2, and HtsH3 to bind to the *lafB* promoter, using an electrophoretic mobility shift assay (EMSA). A PCR-amplified 590 bp DNA fragment from the p*lafB* was used as a probe. The addition of purified HtsH1C-Flag-His, HtsH2C-HA-His, and HtsH3C-Myc-His protein, ranging from 0 to 16 µg, to the reaction mixtures (20 µL and at 28 °C, 25 min) caused a shift in the mobility of the p*lafB* DNA fragment, and the EMSA revealed strong HtsH1, HtsH2, and HtsH3 binding with

the p*lafB* probe in a dose-dependent manner (Fig. 6b–d and Supplementary Fig. 12). We quantified the binding affinity of HtsH1, HtsH2, and HtsH3 to the HSAF operon promoter. In an SPR analysis, HtsH1C-Flag-His directly bound to the promoter of the HSAF biosynthesis gene (p*lafB*) with high affinity ($K_D = 0.5356\,\mu M$) (Fig. 6e). In addition, HtsH2C-HA-His and HtsH3C-Myc-His bound to p*lafB* with $K_D$ values of 1.379 µM and 0.2491 µM, respectively (Fig. 6f, g). The results demonstrated that HtsH1, HtsH2, and HtsH3 could directly target the promoters of the HSAF biosynthesis gene.

Based on the above results, we compared the transcriptome profiles of the wild-type strain and the *htsHs* mutants (Δ*htsH1*, Δ*htsH2*, Δ*htsH3*, Δ*htsH12*, Δ*htsH13*, Δ*htsH23*, and Δ*htsH123*) by RNA-Seq and observed changes in the expression levels of several hundred genes (Supplementary Data 1). We then performed trend analysis of the differential gene expression and found that the amounts of HSAF biosynthesis gene cluster mRNA were constitutively decreased in the *htsHs* mutants (Δ*htsH1*, Δ*htsH2*, Δ*htsH3*, Δ*htsH12*, Δ*htsH13*, Δ*htsH23* and Δ*htsH123*) (Supplementary Fig. 7). Using quantitative RT-PCR (qRT-PCR), we measured the mRNA abundance of *lafB* in the *htsHs* mutants (Δ*htsH1*, Δ*htsH2*, Δ*htsH3*, Δ*htsH12*, Δ*htsH13*, Δ*htsH23*, and Δ*htsH123*), and found that it was reduced compared to that in the wild type (Supplementary Fig. 8).

Taken together, these results suggested that HtsH1, HtsH2, and HtsH3 can directly target the promoters of the HSAF biosynthesis genes to increase their expression and HSAF production by *L. enzymogenes*.

**Phosphorylated HtsH1, HtsH2, and HtsH3 positively regulate HSAF biosynthesis.** Since HtsH1, HtsH2, and HtsH3 function as a group of HyTCS proteins, we explored whether they could directly target p*lafB* depending on its phosphorylation activity in *L. enzymogenes*. To achieve this goal, we assessed the phosphorylation levels of these proteins with calf intestine alkaline phosphatase (CIAP) in the reaction mixtures (20 µL, 28 °C, 60 min). Using $Mn^{2+}$-Phos-tag SDS-PAGE, we showed that the phosphorylation levels of HtsH1C-Flag-His, HtsH2C-HA-His, and HtsH3C-Myc-His decreased upon CIAP treatment (Fig. 7a–c and Supplementary Fig. 12). To test the effect of HtsH1-mediated, HtsH2-mediated, or HtsH3-mediated phosphorylation on the function of the target p*lafB*, EMSAs were performed with the same experimental conditions. The EMSAs revealed that the amount of probes bound to HtsH1C-Flag-His, HtsH2C-HA-His, and HtsH3C-Myc-His decreased with increasing amounts of CIAP (Fig. 7d–f and Supplementary Fig. 12). The above findings indicated that HtsH1, HtsH2 and HtsH3 could directly regulate HSAF biosynthesis gene expression depending on their phosphorylation activity in *L. enzymogenes*.

**RpfG-dependent and HtsH1-dependent, HtsH2-dependent, and HtsH3-dependent regulatory patterns are present in a wide range of bacterial species.** By BLAST analysis of the non-redundant protein sequence (Nr) database of the National Center for Biotechnology Information (NCBI), we found that RpfG, the *rpf* cluster, and HtsH1, HtsH2, and HtsH3 were present not only in the genomes of *Lysobacter* species but also those of *Xanthomonas* species (Supplementary Fig. 9). These findings suggest that RpfG-dependent, HtsH1-dependent, HtsH2-dependent, and HtsH3-dependent regulatory patterns are conserved mechanisms in *Lysobacter* and *Xanthomonas*.

## Discussion

Quorum sensing (QS) allows populations of bacteria to communicate via the exchange of chemical signals, resulting in

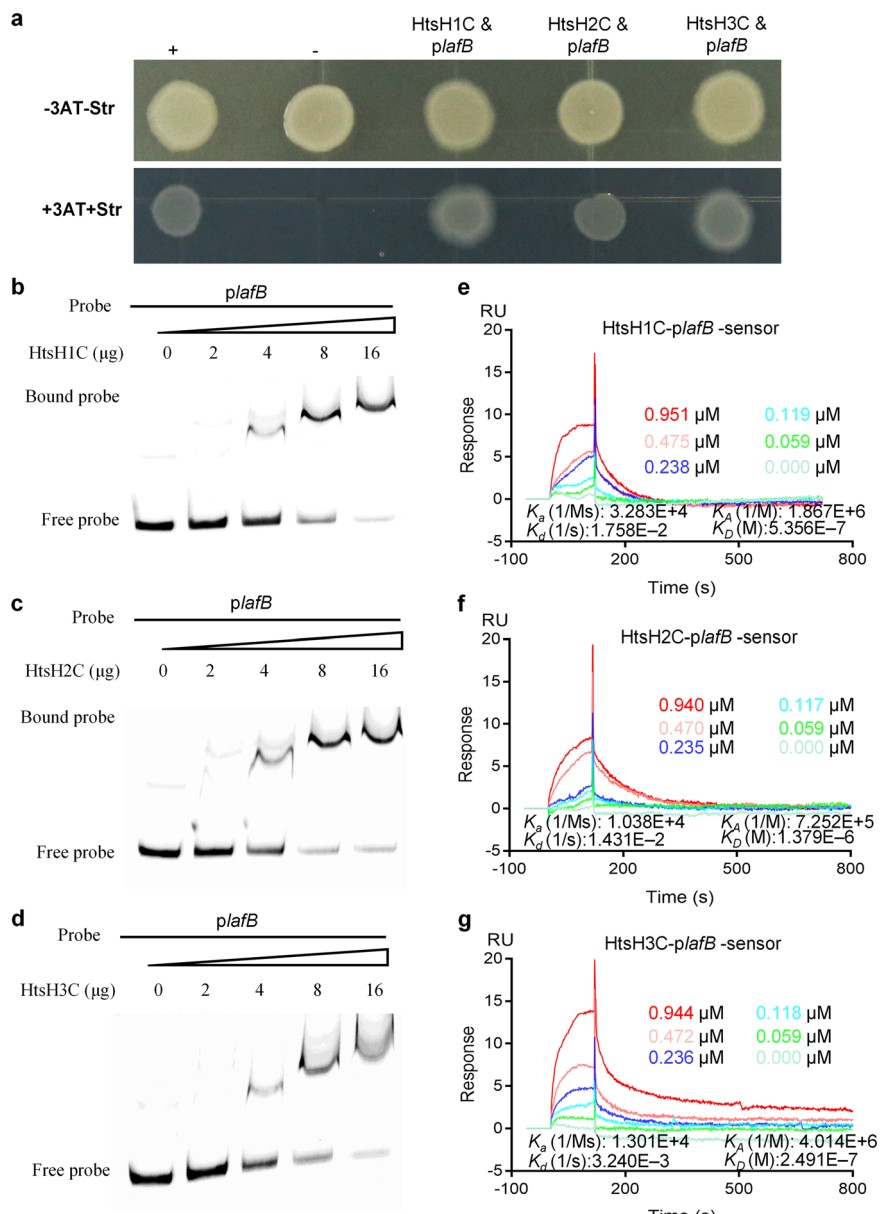

**Fig. 6 HtsH1, HtsH2, and HtsH3 directly bound the promoter of the HSAF biosynthesis gene (p*lafB*). a** Direct physical interaction between HtsH1, HtsH2, and HtsH3 and the *lafB* promoter region was detected in *E. coli*. Experiments were performed according to the procedures described in the "Methods" section. +, cotransformant containing pBX-R2031 and pTRG-R3133, used as a positive control; −, cotransformant containing pBXcmT and the empty pTRG, used as a negative control; HtsH1C and p*lafB*, cotransformant harboring pTRG-HtsH1C and pBXcmT-p*lafB*; HtsH2C and p*lafB*, cotransformant harboring pTRG-HtsH2C and pBXcmT-p*lafB*; HtsH3C and p*lafB*, cotransformant harboring pTRG-HtsH3C and pBXcmT-p*lafB*. -3AT-Str, plate with no selective medium (3AT 3-amino-1,2,4-triazole and Str streptomycin) and +3AT + Str, plate with M9-based selective medium. **b–d** Gel shift assay showing that HtsH1, HtsH2, and HtsH3 directly regulate an HSAF biosynthesis gene. HtsH1C-Flag-His, HtsH2C-HA-His, or HtsH3C-Myc-His protein (0, 2, 4, 8 or 16 μg) was added to reaction mixtures containing 50 ng of probe DNA, and the reaction mixtures were separated on polyacrylamide gels. **e** SPR showing that HtsH1C-Flag-His forms a complex with p*lafB* with $K_D = 0.5356$ μM. **f** SPR showing that HtsH2C-HA-His forms a complex with p*lafB* with $K_D = 1.379$ μM. **g** SPR showing that HtsH3C-Myc-His forms a complex with p*lafB* with $K_D = 0.2491$ μM.

coordinated gene expression in response to cell density[26]. AHL signaling was first discovered in the marine bacterium *Vibrio fischeri*. In *V. fischeri*, LuxI and LuxR produce and respond to 3OC6-HSL, respectively[27,28]. In addition to *V. fischeri*, *Pseudomonas aeruginosa* has emerged as an important model organism for QS research[27]. In *P. aeruginosa*, the LasIR system produces and responds to 3OC12-HSL, and the RhlR system produces and responds to C4-HSL[26,27]. QscR is an orphan LuxR receptor that is not linked to a luxI synthase gene. QscR responds to 3OC12-HSL produced by LasI[27,29]. The quorum-sensing transcriptional

activator TraR of *Agrobacterium tumefaciens*, which controls the replication and conjugal transfer of the tumor-inducing (Ti) virulence plasmid, is inhibited by the TraM antiactivator[30,31]. In the DSF-mediated quorum sensing system, RpfC undergoes autophosphorylation upon sensing accumulated extracellular DSF signals. Through the conserved phosphorelay mechanism, RpfG is phosphorylated, which leads to activation of its c-di-GMP phosphodiesterase activity. Degradation of c-di-GMP releases Clp, which regulates subsets of virulence genes directly or through the downstream transcription factors FhrR and Zur[5,11].

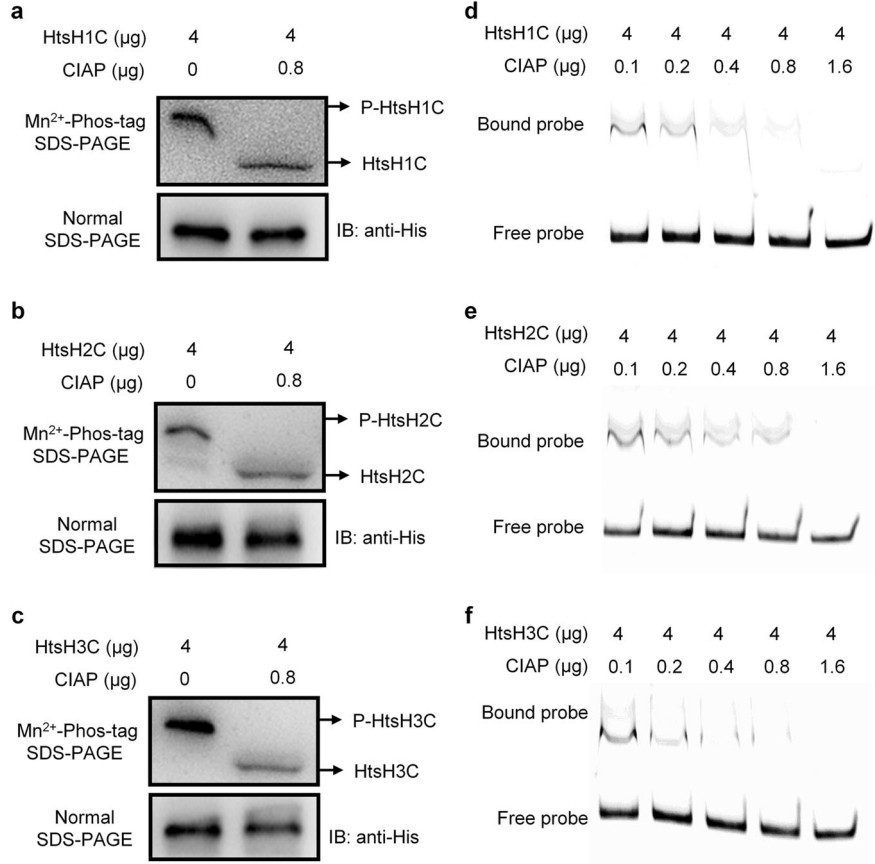

**Fig. 7 HtsH1, HtsH2, and HtsH3 directly target plafB depending on their PDE activity. a–c** Phosphorylation analysis of HtsH1C-Flag-His, HtsH2C-HA-His, or HtsH3C-Myc-His with calf intestine alkaline phosphatase (CIAP) at concentrations ranging from 0.1 to 1.6 µg in the reaction mixtures (20 µL, 28 °C, 60 min) using $Mn^{2+}$-Phos-tag SDS-PAGE. **d–f** Gel shift assay showing that HtsH1C-Flag-His, HtsH2C-HA-His, or HtsH3C-Myc-His does not directly target p*lafB* with calf intestine alkaline phosphatase (CIAP) at concentrations ranging from 0.1 to 1.6 µg in the reaction mixtures (20 µL, 28 °C, 60 min). HtsHs (4 µg) were added to reaction mixtures containing 50 ng of probe DNA, and the reaction mixtures were separated on polyacrylamide gels.

In this study, we revealed a signal pathway by which the DSF type-based QS system component protein RpfG to interacts with HtsH1, HtsH2, and HtsH3 to regulate the biosynthesis of HSAF in *L. enzymogenes*.

In previous studies, we and our collaborators have shown that QS was employed by *L. enzymogenes* to affect production of the antifungal factor HSAF[6,16]. However, the mechanism through which QS coordinates the synthesis of HSAF remains unknown. RpfG, as a necessary component protein of the DSF mediated QS signal transduction system, contains a C-terminal HD-GYP domain that can affect production of the antifungal factor HSAF in *L. enzymogenes*[12,16]. However, how RpfG regulates the synthesis of HSAF remains incompletely studied. The results of the present study provide biochemical, genetic, and biophysical evidence to demonstrate that *L. enzymogenes* reprograms the QS signal system that RpfG interacts with HtsHs to regulate the biosynthesis of the antifungal antibiotic HSAF. HtsH1, HtsH2, and HtsH3 regulate the expression of the synthetic genes of the antifungal factor HSAF depending on their phosphorylation (Fig. 8).

HD-GYP domain proteins are c-di-GMP PDEs that can degrade c-di-GMP[13]. However, the role of the HD-GYP domain of RpfG in the degradation of c-di-GMP in *L. enzymogenes* has remained unelucidated. Therefore, we tested the PDE activity of RpfG and successfully showed that it was able to degrade the model substrate c-di-GMP to 5′-pGpG. To test whether the HD-GYP motif was important for catalytic activity in RpfG, we constructed RpfG mutant proteins (RpfG-H190A-MBP, RpfG-

D191A-MBP, RpfG-G253A-MBP, and RpfG-P255A-MBP). We tested the c-di-GMP PDE activity of these mutant proteins and suggested that the HD-GYP domain was required for full PDE activity of RpfG in vivo. It is generally speculated that the PDE activity of HD-GYP domain proteins degrades c-di-GMP to GMP[12,32,33]. However, we showed that the activity of the HD-GYP domain of RpfG is involved in the degradation of c-di-GMP to 5′-pGpG.

Intriguingly, we found that the strains expressing the RpfG H190A, D191A, G253A, Y254A, and P255A mutant proteins resulted in increased HSAF production compared with the Δ*rpfG* mutant strain. Importantly, we found that the Δ*rpfG* mutant did not significantly change c-di-GMP production compared with of the wild-type strain in the antifungal factor HSAF-production medium (10% TSB). These data demonstrated that the regulatory activity of RpfG does not depend on its PDE enzymatic activity. This is the first report showing that a PDE does not depend on its c-di-GMP-degrading activity to regulate a downstream pathway. Thus, we wondered whether RpfG regulates HSAF synthesis through interactions with other proteins in *L. enzymogenes*.

Bioinformatics predictions have shown that RpfG may interact with three HyTCS proteins (HtsH1, HtsH2, and HtsH3). Then, we used pull-down and SPR to demonstrate the binding events between the RpfG and HtsH1, HtsH2, or HtsH3 proteins. Notably, RpfG and HtsH1, HtsH2, or HtsH3 interactions affect the PDE activity of RpfG. However, how RpfG affects HtsH1, HtsH2, or HtsH3 remains unknown. We speculate that RpfG may affect HtsH1, HtsH2, or HtsH3 autophosphorylation. However,

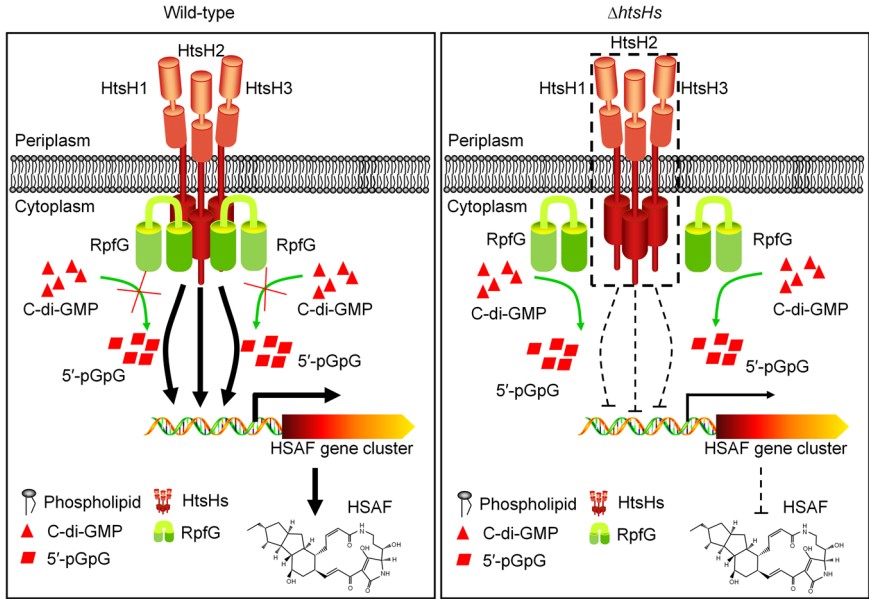

**Fig. 8 Schematic of the proposed RpfG directly interacting with three HyTCS proteins (HtsH1, HtsH2, and HtsH3) to regulate HSAF biosynthesis.** The potential regulatory pathways and interactions of RpfG with HtsH1, HtsH2, and HtsH3 are proposed according to our observations and previous studies. RpfG and HtsH1, HtsH2, or HtsH3 interactions affect the PDE activity of RpfG. Phosphorylated HtsH1, HtsH2, and HtsH3 can directly target the promoter of HSAF biosynthesis genes to regulate HSAF production in *L. enzymogenes*. The solid arrow indicates the demonstrated direct signal modulation. The dashed arrow suggests a potential signal regulation pathway. The dashed box represents the deletion of the *htsH1*, *htsH2*, and *htsH3* genes.

we could not obtain the full-length HtsH1, HtsH2, and HtsH3 proteins, so further clarification of these possible mechanisms will help elucidate the mechanism underlying the RpfG interaction with HtsH1, HtsH2, or HtsH3. Moreover, we found that *htsH1*, *htsH2*, and *htsH3* likely constitute a single transcription unit. HyTCS-based regulation may be crucial for responding to environmental changes and finely tuning gene expression[34–37]. However, the biological function of three consecutive HyTCS proteins has not been reported in bacteria.

In this study, we found that the in-frame deletion of the *htsH1*, *htsH2*, and *htsH3* coding sequences significantly decreased HSAF production. Thus, we speculate that RpfG may interact with three HyTCS proteins to coordinate HSAF production in *L. enzymogenes*. To test this hypothesis, we performed an *E. coli*-based one-hybrid assay and EMSA. The results demonstrated that HtsH1, HtsH2, and HtsH3 could directly target the promoters of HSAF biosynthesis genes. We further analysed the transcription level of HSAF biosynthesis-related genes in *htsH1*, *htsH2*, and *htsH3* mutants. Knockout of *htsH1*, *htsH2*, and *htsH3* significantly reduced the transcription level of the antifungal factor HSAF biosynthesis genes. These results suggest that HtsH1, HtsH2, and HtsH3 can directly regulate HSAF biosynthesis gene expression and increase production of the antifungal factor HSAF in *L. enzymogenes*. In this study, we found that RpfG interacts with the HtsH1, HtsH2, or HtsH3 protein and that HtsH1, HtsH2, and HtsH3 can interact with each other. We also found that knockout of *rpfG* or *htsH1*, *htsH2*, and *htsH3* significantly reduced HSAF production. These results suggest that RpfG forms a complex with HtsH1, HtsH2 and HtsH3 to regulate HSAF biosynthesis. However, the mechanism needs to be further elucidated.

Phosphorylation of TCS is critical for regulating the expression of downstream genes[38–40]. Phosphorylated HtsH1, HtsH2, and HtsH3 positively regulate HSAF biosynthesis and argue with the existing literature concerning whether this is a common route of gene expression regulation. We used Mn²⁺-Phos-tag SDS-PAGE and EMSA to show that HtsH1, HtsH2, and HtsH3 target p*lafB* depending on their phosphorylation activity. In this study, we report the biological functions of the three HyTCS proteins

HtsH1, HtsH2, and HtsH3 in the regulation of antibiotic biosynthesis.

One of the notable results of this study is that RpfG, and HtsH1, HtsH2, and HtsH3 regulatory patterns seem to be conserved mechanisms in *Lysobacter* and *Xanthomonas*. To our knowledge, RpfG represents a unique example of a c-di-GMP metabolic enzyme that directly interacts with three HyTCS proteins (HtsH1, HtsH2, and HtsH3) to regulate HSAF biosynthesis.

## Methods

**Bacterial strains, plasmids, and growth conditions.** The strains and plasmids used in this study are shown in Supplementary Table 1. *E. coli* strains were grown in Luria-Bertani medium (10 g/L tryptone, 5 g/L yeast extract, 10 g/L NaCl, pH 7.0) at 37 °C. *L. enzymogenes* strains were grown at 28 °C in Luria-Bertani medium and 10% TSB. For the preparation of culture media, tryptone, peptone, beef extract, and yeast extract were purchased from Sangon Biotech (Shanghai, China). When required, antibiotics were added (30 μg/mL kanamycin sulfate, 50 μg/mL gentamycin) to the *E. coli* or *L. enzymogenes* cultures. The bacterial growth in liquid medium was determined by measuring the optical density at 600 nm (OD600) using a Bioscreen-C Automated Growth Curves Analysis System (OY Growth Curves FP-1100-C, Helsinki, Finland).

**Site-directed mutagenesis.** Site-directed mutagenesis and essentiality testing were performed as follows[41]. To obtain the RpfG mutant proteins and *rpfG* site-directed mutant strains, plasmids harboring mutations in *rpfG* were constructed. For example, to obtain the H190A mutation in RpfG, approximately 500-bp DNA fragments flanking the *rpfG* gene were amplified with *Pfu* DNA polymerase using *L. enzymogenes* genomic DNA as template and either MBP-*rpfG* P1 and *rpfG* H190A P1 (for the Up *rpfG* H190A mutant), or *rpfG* H190A P1 and MBP-*rpfG* P2 (for the Down *rpfG* H190A mutant) as the primers (Supplementary Table 2). The fragments were connected by overlap PCR using the primers MBP-*rpfG* P1 and MBP-*rpfG* P2. The fused fragment was digested with *Bam*H I and *Hin*dIII and inserted into pMAl-p2x to obtain the plasmid pMAl-*rpfG* H190A. The other four site-directed mutant plasmids (D191A, G253A, Y254A, and P255A) were constructed using a similar method.

**Protein expression and purification.** Protein expression and purification were performed as follows[42]. To clone the *rpfG* gene, genomic DNA extracted from *L. enzymogenes* was used for PCR amplification using *Pfu* DNA polymerase, and the primers are listed in Supplementary Table 2. The PCR products were inserted into pMAl-p2x to produce the plasmids pMAl-*rpfG*. The *rpfG* gene was verified by nucleotide sequencing by Genscript (Nanjing, Jiangsu, China). Le *rpfG* and *rpfG* site-directed mutants with a vector-encoded maltose binding protein in the

N-terminus were expressed in *E. coli* BL21 (DE3) and purified with Dextrin Sepharose High Performance (Qiagen, Chatsworth, CA, USA) using an affinity column (Qiagen). The protein purity was monitored by SDS-PAGE. His$_6$-tagged protein expression and purification were performed as described previously[41–43].

**PDE activity assays in vitro.** The PDE activity assay was performed essentially as follows[25]. Briefly, 2 μM MBP-RpfG or its derivatives were tested in buffer containing 60 mM Tris-HCl (pH 7.6), 50 mM NaCl, 10 mM MnCl$_2$, and 10 mM MgCl$_2$. The reaction was started by the addition of 100 μM c-di-GMP. All reaction mixtures were incubated at 28 °C for 5–60 min, followed by boiling for 10 min to stop the reaction. The samples were filtered through a 0.2 μM pore size cellulose-acetate filter, and 20 μL of each sample was loaded onto a reverse-phase C18 column and separated by HPLC. The separation protocol involved two mobile phases, 100 mM KH$_2$PO$_4$ plus 4 mM tetrabutylammonium sulfate (A) and 75% A + 25% methanol (B).

**C-di-GMP extraction and quantification.** C-di-GMP extraction and quantification were performed as follows[25]. Cultures were grown in 10% TSB at 28 °C until the OD600 reached 1.5 based on the growth curve. Cells from 2 mL of the culture were harvested for protein quantification by BCA (TransGen, China). Cells from 8 mL of culture were used for c-di-GMP extraction using 0.6 M HClO$_4$ and 2.5 M K$_2$CO$_3$. The samples were subjected to 0.22 μm Mini-Star filtration, and the filtrate was concentrated to 100 μL for liquid chromatography-tandem mass spectrometry (LC-MS/MS) analysis on an AB SCIEX QTRAP 6500 LC-MS/MS system (AB SCIEX, USA). The separation protocol involved two mobile phases, buffer A: 100 mM ammonium acetate plus 0.1% acetic acid; buffer B: 100% methanol. The gradient system was from 90% buffer A and 10% buffer B to 20% buffer A and 80% buffer B. The running time was 9 min, and the flow rate was 0.3 mL/min.

**Gene deletion and complementation.** The in-frame deletions in *L. enzymogenes* OH11 were generated via double-crossover homologous recombination[25,41] using the primers listed in Supplementary Table 2. In brief, the flanking regions of each gene were PCR-amplified and cloned into the suicide vector pEX18Gm (Supplementary Table 1). The deletion constructs were transformed into the wild-type strain by electroporation, and gentamycin was used to select for integration of the nonreplicating plasmid into the recipient chromosome. A single-crossover integrant colony was spread on LB medium without gentamycin and incubated at 28 °C for 3 days, and after appropriate dilution, the culture was spread on LB plates containing 15% sucrose. Colonies sensitive to gentamycin were screened by PCR using the primers listed in Supplementary Table 2, and the gene deletion strains were obtained.

For gene complementation constructs, DNA fragments containing the full-length genes along with their promoters were PCR amplified and cloned into the versatile plasmid pBBR1MCS5[44]. The resulting plasmids were transferred into the *L. enzymogenes* strain by electroporation, and the transformants were selected on LB plates containing Gm.

**RNA-Seq.** The RNA-Seq assay was performed as follows[45,46]. Briefly, the wild-type, ΔhtsH1, ΔhtsH2, ΔhtsH3, ΔhtsH12, ΔhtsH13, ΔhtsH23, and ΔhtsH123 mutant strains were grown in 10% TSB medium at 28 °C, and their cells were collected when the OD600 reached 1.0 based on the growth curve. The collected cells were used for RNA extraction by the TRIzol-based method (Life Technologies, CA, USA), and RNA degradation and contamination were monitored on 1% agarose gels. Then, clustering and sequencing were performed by Genedenovo Biotechnology Co., Ltd (Guangzhou, Guangdong, China). To analyse the DEGs between the wild-type, ΔhtsH1, ΔhtsH2, ΔhtsH3, ΔhtsH12, ΔhtsH13, ΔhtsH23, and ΔhtsH123 mutant strains, the gene expression levels were further normalized using the fragments per kilobase of transcript per million (FPKM) mapped reads method to eliminate the influence of different gene lengths and amounts of sequencing data on the calculation of gene expression. The edgeR package (http://www.r-project.org/) was used to determine DEGs across samples with fold changes ≥2 and a false discovery rate-adjusted *P* (*q*-value) < 0.05. DEGs were then subjected to enrichment analysis of GO functions and KEGG pathways, and *q* values were corrected using <0.05 as the threshold.

**Quantitative real-time PCR.** The bacterial cells were collected when the cellular optical density (OD600) reached 1.0 in 10% TSB. Total RNA was extracted using a TRIzol-based method (Life Technologies, CA, USA). RNA quality control was performed via several steps: (1) the degree of RNA degradation and potential contamination were monitored on 1% agarose gels; (2) the RNA purity (OD260/OD280, OD260/OD230) was checked using a NanoPhotometer® spectro-photometer (IMPLEN, CA, USA); and (3) the RNA integrity was measured using a Bioanalyzer 2100 (Agilent, Santa Clara, CA, USA). The primers used in this assay are listed in Supplementary Table 2. cDNA was then synthesized from each RNA sample (400 ng) using the TransScript® All-in-One First-Strand cDNA Synthesis SuperMix for qPCR (One-Step gDNA Removal) Kit (TransGen Biotech, Beijing, China) according to the manufacturer's instructions. qRT-PCR was performed using TransStart Top Green qPCR SuperMix (TransGen Biotech) on a Quant-Studio TM 6 Flex Real-Time PCR System (Applied Biosystems, Foster City, CA,

USA) with the following thermal cycling parameters: denaturation at 94 °C for 30 s, followed by 40 cycles of 94 °C for 5 s and 60 °C for 34 s. Gene expression analyses were performed using the $2^{-\Delta\Delta CT}$ method with 16S rRNA as the endogenous control, and the expression level in the wild type was set to a value of 1. The experiments were performed three times, and three replicates were examined in each run.

**Pull-down assay.** The assay was performed as follows[25]. Briefly, the purified proteins were used to perform the pull-down assay in a reaction system comprising 800 μL PBS buffer, 5 μM (final concentration) MBP-RpfG and HtsH1C-Flag-His, HtsH2C-HA-His or HtsH3C-Myc-His proteins, and 50 μL Dextrin Sepharose High Performance (Sigma-Aldrich, St. Louis, MO, USA). All samples were incubated at 4 °C overnight. The agarose was collected by centrifugation and washed ten times with PBS containing 1% Triton X-100 to remove non-specifically bound proteins. The MBP-bead-captured proteins were eluted by boiling in 6× SDS loading dye for 10 min. These samples were subjected to SDS-PAGE and Western blotting. Protein detection involved the use of MBP-specific (ab49923), Flag-specific (ab1162), HA-specific (ab187915), Myc-specific (ab32072), and His-specific (ab18184) antibodies obtained from Abcam, UK.

**Phosphorylation analysis through Phos-tag gel.** The purified HtsH1C-Flag-His, HtsH2C-HA-His, or HtsH3C-Myc-His proteins (100 ng) were incubated with CIAP (Solarbio, Beijing, China) at 28 °C for 60 min and resolved by 8% SDS-PAGE prepared with 50 μM acrylamide-dependent Phos-tag ligand and 100 μM MnCl$_2$[47]. Gel electrophoresis was performed with a constant voltage of 80 V for 3–6 h. Before transfer, the gels were equilibrated in transfer buffer with 5 mM EDTA for 20 min two times, followed by transfer buffer without EDTA for another 20 min. Protein transfer from the Mn$^{2+}$ phos-tag acrylamide (APExBIO, Houston, USA) gel to the PVDF membrane (Millipore, Massachusetts, USA) was performed for ~24 h at 80 V at 4 °C, and then the membrane was analysed by Western blotting using the anti-His antibody.

**Bacterial one-hybrid assays.** Bacterial one-hybrid assays were performed as follows[48,49]. In brief, the bacterial one-hybrid reporter system contains three components: the plasmids pBXcmT and pTRG, which are used to clone the target DNA and to express the target protein, respectively, and the *E. coli* XL1-Blue MRF′ kan strain, which is the host strain for the propagation of the pBXcmT and pTRG recombinants[50]. In this study, the promoter of the HSAF biosynthesis gene (p*lafB*) was cloned into pBXcmT to generate the recombinant vector pBXcmT-p*lafB*. Similarly, the coding regions of Le *htsH1*, Le *htsH2*, and Le *htsH3* were cloned into pTRG to create the final constructs pTRG-*htsH1*, pTRG-*htsH2*, and pTRG-*htsH3*, respectively. The two recombinant vectors were transformed into the XL1-Blue MRF′ kan strain. If direct physical binding occurred between HtsH1, HtsH2, or HtsH3 and p*lafB*, the positive-transformant *E. coli* strain containing both pBXcmT-p*lafB* and pTRG-HtsHs would grow well on selective medium, that is, minimal medium containing 5 mM 3-amino-1,2,4-triazole, 8 μg/mL streptomycin, 12.5 μg/mL tetracycline, 34 μg/mL chloramphenicol, and 30 μg/mL kanamycin. Furthermore, cotransformants containing pBX-R2031/pTRG-R3133 served as a positive control[50], and cotransformants containing either empty pTRG or pBXcmT-p*lafB* were used as negative controls. All cotransformants were spotted onto selective medium, grown at 28 °C for 3–4 days, and then photographed.

**Electrophoretic mobility gel shift assays (EMSAs).** Electrophoretic mobility gel shift assays were performed as follows[51,52]. For HtsH1, HtsH2, or HtsH3 gel shift assays, we used DNA fragments that included p*lafB* as a probe. The probe DNA (50 ng) was mixed with protein in a 20 μL reaction mixture containing 10 mM Tris-HCl (pH 7.5), 50 mM KCl, 1 mM dithiothreitol, and 0.4% glycerol. After incubation for 30 min at 28 °C, samples were electrophoresed on a 5% non-denaturing acrylamide gel in 0.5× TBE buffer at 4 °C. The gel was soaked in 10,000-fold-diluted SYBR Green I nucleic acid dye (Sangon Biotech, Shanghai, China), and the DNA was visualized at 300 nm.

**HSAF extraction and quantification.** HSAF extraction and quantification were performed as follows[53]. HAF was extracted from 4 mL *L. enzymogenes* cultures grown in 10% TSB for 48 h at 28 °C with shaking (at 180 rpm) and adjusted to pH 2.5 with HCl. Same volume of ethyl acetate was added to the acidified broth and the mixture was shaken in a vortexer at 2000 rpm for 1 min. The ethyl acetate fractions were collected, and ventilated to dryness in a fume hood. The residue was dissolved in 100 μL of methanol. The crude extract was subjected to 0.22 μm Mini-star filtration, and the filtrate was concentrated to 100 μL. The extract (20 μL) was used for high-performance liquid chromatography (HPLC) analysis using a C18 reversed-phase HPLC column (4.6 × 250 mm, Agilent Technologies, Inc.) with detection at 318 nm. Pure water and acetonitrile containing 0.04% (v/v) Tri-fluoroacetic acid (TFA) were used as the A and B mobile phases, respectively. The gradient program used a flow rate of 1 mlL/min.

**Statistics and reproducibility.** The experimental datasets were subjected to analyses of variance using GraphPad Prism 7.0. The significance of the treatment

effects was determined by the $F$ value ($P = 0.05$). If a significant $F$ value was obtained, separation of means was accomplished by Fisher's protected least significant difference at $P \leq 0.05$.

**Reporting summary**. Further information on research design is available in the Nature Research Reporting Summary linked to this article.

## Data availability

The data that support the findings of this study are openly available in GenBank at https://www.ncbi.nlm.nih.gov/nuccore/, accession numbers RCTY01000033 (*Lysobacter enzymogenes* strain OH11 scffold34, whole genome shotgun sequence; Le4727/Le RpfG, locus tag = D9T17_13845), RCTY01000055 (*Lysobacter enzymogenes* strain OH11 scffold56, whole genome shotgun sequence, Le3071/Le HtsH1, locus tag = D9T17_21400)), RCTY01000054 (*Lysobacter enzymogenes* strain OH11 scffold55, whole genome shotgun sequence, Le3072/Le HtsH2, locus tag = D9T17_21390), RCTY01000054 (*Lysobacter enzymogenes* strain OH11 scffold55, whole genome shotgun sequence, Le3073/Le HtsH3, locus tag = D9T17_21385). RNA-sequencing raw data are deposited into the NCBI's Sequence Read Archive (SRA) and are accessible through BioProject series accession number PRJNA758119. All source data to generate graphs have been combined in Supplementary Data 2. Any remaining data are available from the corresponding author on reasonable request.

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

## Acknowledgements

The authors thank Jialei Wang, Ning Wang, and Cunfa Xu at the Central Laboratory of Jiangsu Academy of Agricultural Sciences for LC-MS/MS and SPR technical support.

This study was funded by the National Natural Science Foundation of China (31872018), the Natural Science Foundation of Jiangsu Province (BK20190266), and the China Agriculture Research System of MOF and MARA.

## Author contributions

K.L. and F.L. conceived and designed the experiments. K.L., G.X., B.W., R.H., and G.W. performed the experiments. K.L. analysed the data and prepared the figures. K.L. wrote the manuscript draft. F.L. revised the manuscript. All authors read and approved the final manuscript.

## Competing interests

The authors declare no competing interests.
