## [Peer Review File · Communications Biology]

The predatory soil bacterium *Lysobacter* reprograms quorum sensing system to regulate antifungal antibiotic production in a cyclic-di-GMP-independent mannerReviewers' comments:

Reviewer #1 (Remarks to the Author):

Li et al. describe the characterization of the hybrid two-component system involved in HSAF production. The design and experiments were well done to support the results and conclusions of this manuscript. The novel information about this TCS will improve our understanding about bacterial QS systems. Therefore, I am willing to accept this manuscript for the publication in Communication Biology. Before that, I recommend the authors to revise the following things.

1. Line 78&Fig 1A. The authors expressed RpfG proteins and checked their purity by SDS-PAGE. However, the results indicated that the samples contains several similar size proteins. The authors should mentioned these impurities (or decomposition of RpfG) in the results or method sections.
2. Line 114&Fig 1C. I understand the results of c-di-GMP concentrations. If possible, I would like to know the dynamics of c-di-GMP concentrations (e.g., 0-60 h). The data will exclude the possibility that the PDF enzymatic activity can change the stage of cell growth.
3. Line 128. I cannot fully follow the previous information about the HyTCS protein HtsH. I recommend the authors to add some additional information in results (or introduction).

Reviewer #2 (Remarks to the Author):

- What are the major claims of the paper?

That the QS-associated gene RpfG actually has a secondary role from its activity as a PDE to cleave c-di-GMP. Rather in this work RpfG appears to bind to the HtsHS1-3 proteins and regulate the production of HSAF an antimicrobial. It suggests an alternate mechanism for regulating this process.

- Are the claims novel? If not, please identify the major papers that compromise novelty.

The claims are generally novel. I have some concern that I found a nearly identical manuscript here as a pre-print: <https://www.biorxiv.org/content/10.1101/2020.07.13.201541v1.full>
This other article does not make as much of a case for the QS reprogramming.

- Will the paper be of interest to others in the field?

The discovery of novel strategies for how QS circuits function are of particular interest.

- Will the paper influence thinking in the field?

o The observation that the relevant proteins may be broadly distributed in soil microorganisms may broaden the search for the same, or similar, regulatory mechanisms for HSAF production as well as other responses.

- Are the claims convincing? If not, what further evidence is needed?

o The authors state that the PDE activity of the c-terminus is known but then proceed to evaluate it. Since the PDE-activity was not in doubt the experiment seems unnecessary. A better explanation for the need for this study should be made

o The model in Figure 8 doesn't do a good job to explain the model, neither does the figure caption. Is the thickness of the lines intended to reflect the amount of ligand available?

o It is unusual to have a membrane bound protein bind a transcription factor directly as proposed here. While the data seems reasonable it seems like it certainly deserves increased discussion

o The studies seem disconnected from the role of the system in actual QS. How this system responds in the presence of DSF is critical to understanding how this functions in soil.

o Similarly, how does RpfG association with these other proteins influence DSF perception? A better explanation of the QS circuit they are asserting this operates through needs to be presented.

- Are there other experiments that would strengthen the paper further? How much would they improve it, and how difficult are they likely to be?

o The exogenous addition of DSF to evaluate its effects on HSAF production or a reference to this work previously done elsewhere would do a better job of framing how this process works in the context of the role of QS normally held by RpfG. This should not be difficult.

o How does the presence of DSF influence the association between RpfG and HtsH1-3?

- Are the claims appropriately discussed in the context of previous literature?

o This is the co-opting/reprogramming of an existing QS circuit to function in a decidedly non-QS dependent manner. This departure needs further discussion. There are lots of permutations to QS circuits such as LasR/QscR, TraM and other unusual regulators. This seems unique in being hijacked from the QS strategies. It really needs a better discussion in context of how their results impacts our understanding of QS.

- If the manuscript is unacceptable in its present form, does the study seem sufficiently promising that the authors should be encouraged to consider a resubmission in the future?

o While a better discussion, and possibly additional experiments, are required to address the connection to QS, they should be encouraged to resubmit.

Reviewer #3 (Remarks to the Author):

This paper, for the first time, studied three hybrid two-component system (HyTCS) proteins, HtsH1, HtsH2, and HtsH3 in *Lysobacter*, and showed many interesting results. RpfG, the sensor protein with PDE activity in DSF mediated QS system, can interact directly with HtsH. The authors also provide convincing evidence that the phosphorylated HtsHs can interact with promoter of the HSAF synthetic cluster to regulate HSAF amount, while htsHs gene deletion cause no effects on growth.

Specific comments:

1. The authors showed the evidence that all the three htsHs are in one transcript, indicating the only promoter located upstream of htsH1. The author also constructed a serials of htsHs mutants, then the expression of htsH2 or htsH3 may preserve or be affected in htsH1 deletion mutant. The author should clarify that expression of htsHs downstream preserved.

2. All the three HtsHs are membrane-bounded proteins, as shown in Fig. 8, while the authors showed they bind to the promoter of HSAF synthetic gene cluster directly. Is there other examples that membrane-bounded proteins can interact with promoter ? or other intermediate protein transfers the activation or inhibition effect ?

3. The authors confirmed that RpfG binds directly to the HtsHs by pull-down assay, and HtsHs bind to the lafB promoter and positively regulate HSAF production. However, how RpfG affects HtsHs to bind the lafB promoter and positively regulate HSAF production?.

4. HtsHs bind to the lafB promoter, but what DNA sequence in lafB promoter interact with HtsHs directly?

5. The authors showed phosphorylated HtsH1, HtsH2, and HtsH3 positively regulate HSAF biosynthesis, could you show resders which amino acid residue in HtsHs was phosphorylated?

Line 98: the word "of" should be deleted in "the production of in...."

Line 145: "the cytoplasmic fragments of HtsH1, HtsH2, and HtsH3 (HtsH1C-Flag-His, HtsH2C-HA-His, and HtsH3C-Myc-His, respectively", why used three different tag (flag,HA, Myc) fused into the three proteins ?

Line 152: "HtsH1, HtsH2, or HtsH3 proteins"should be clearly marked as HtsH1C....

Line 199: Why choose the first gene lafB in the HSAF biosynthesis operon (PlafB) ? LafB is the key enzyme for HSAF synthesis ? Does HtsHs bind to other promoter in the operon ?

List of Responses

Dear Editors and Reviewers:

Thank you for your letter and for the reviewers' comments concerning our manuscript entitled "A predatory soil bacterium reprograms a quorum sensing signal system to regulate antifungal weapon production in a cyclic-di-GMP-independent manner" (ID: COMMSBIO-21-0536). These comments are all valuable and very helpful for revising and improving our paper and provide important guidance in our research. We have studied the comments carefully and have made corrections that we hope meet with your approval. For convenience, the reviewers' comments are in black, and the responses are in blue. The revised portions are marked in red in the paper. The main corrections in the paper and the responses to the reviewer comments are as follows:

Responses to the reviewers' comments:

Reviewer #1 (Remarks to the Author):

Li et al. describe the characterization of the hybrid two-component system involved in HSAF production. The design and experiments were well done to support the results and conclusions of this manuscript. The novel information about this TCS will improve our understanding about bacterial QS systems. Therefore, I am willing to accept this manuscript for the publication in *Communication Biology*. Before that, I recommend the authors to revise the following things.

Response: Thank you for your advice. We have carefully revised our paper. We hope the revised manuscript meets with your approval.

1. Line 78&Fig 1A. The authors expressed RpfG proteins and checked their purity by SDS-PAGE. However, the results indicated that the samples contains several similar size proteins. The authors should mentioned these impurities (or decomposition of RpfG) in the results or method sections.

Response: Thank you for your comments. We have revised the sentence in lines 81-83 of the manuscript: "The proteins had a monomeric molecular weight of 71 kDa, as observed by SDS gel electrophoresis, and were purified by Dextrin Sepharose High Performance to obtain the preparations (Figure 1A). The RpfG protein was fused with the MBP tag, leading to the presence of some impurities".

2. Line 114&Fig 1C. I understand the results of c-di-GMP concentrations. If possible, I would like to know the dynamics of c-di-GMP concentrations (e.g., 0-60 h). The data will exclude the possibility that the PDF enzymatic activity can change the stage of cell growth.

Response: We thank the reviewer for the suggestion to help us improve this work. Fig. 1C shows the PDE enzyme activity of RpfG and the His-190, Asp-191, Gly-253, and Pro-255 point mutations of RpfG at different time points from 0-30 min. These results suggest that RpfG has PDE activity and that the individual HD-GYP residues are required for full PDE activity of RpfG *in vivo*. We also quantified the intracellular c-di-GMP concentrations in the HSAF-production medium (10% TSB) in 48 h and also found that deletion of *rpfG* did not affect the intracellular c-di-GMP concentrations. The results were similar to those shown in Fig. 2D previously.

3. Line128. I cannot fully follow the previous information about the HyTCS protein HtsH. I recommend the authors to add some additional information in results (or introduction).

Response: Thank you for your insightful comments and suggestions. The biological functions of three consecutive HyTCS proteins have not been reported in bacteria. We report for the first time the biological functions of three consecutive HyTCS. So we designated the HyTCS proteins name as HtsH (Hybrid two-component signalling system regulating HSAF production) based on the findings of this study. In our manuscript, we introduced the information of HtsHs HyTCS protein. Please refer to lines 146-158 of the new manuscript.

Reviewer #2 (Remarks to the Author):

- What are the major claims of the paper?

That the QS-associated gene RpfG actually has a secondary role from its activity as a PDE to cleave c-di-GMP. Rather in this work RpfG appears to bind to the HtsHS1-3 proteins and regulate the production of HSAF an antimicrobial. It suggests an alternate mechanism for regulating this process.

Response: Thank you for your insightful comments and suggestions.

- Are the claims novel? If not, please identify the major papers that compromise novelty.

The claims are generally novel. I have some concern that I found a nearly identical manuscript here as a pre-print: <https://www.biorxiv.org/content/10.1101/2020.07.13.201541v1.full>

This other article does not make as much of a case for the QS reprogramming.

Response: Thank you for your comments.

- Will the paper be of interest to others in the field?

The discovery of novel strategies for how QS circuits function are of particular interest.

Response: Thank you for your kind comments.

- Will the paper influence thinking in the field?

o The observation that the relevant proteins may be broadly distributed in soil microorganisms may broaden the search for the same, or similar, regulatory mechanisms for HSAF production as well as other responses.

Response: Thank you for your kind comments.

• Are the claims convincing? If not, what further evidence is needed?

o The authors state that the PDE activity of the c-terminus is known but then proceed to evaluate it. Since the PDE-activity was not in doubt the experiment seems unnecessary. A better explanation for the need for this study should be made.

Response: Thank you for your kind comments. In the new manuscript, we have added the sentence as follows in lines 81-83 of the new manuscript: “HD-GYP domain-containing proteins can degrade c-di-GMP to GMP and 5'-pGpG. However, the in vitro enzyme activity of RpfG homologues has not been studied and identified”.

o The model in Figure 8 doesn't do a good job to explain the model, neither does the figure caption. Is the thickness of the lines intended to reflect the amount of ligand available?

Response: Thank you for your comments. In the new manuscript, we have revised our statement as follows in lines 821-829 of manuscript: “Figure 8. Schematic of the proposed RpfG directly interacting with three HyTCS proteins (HtsH1, HtsH2, and HtsH3) to regulate HSAF biosynthesis. The potential regulatory pathways and interactions of RpfG with HtsH1, HtsH2, and HtsH3 are proposed according to our observations and previous studies. RpfG and HtsH1, HtsH2, or HtsH3 interactions affect the PDE activity of RpfG. Phosphorylated HtsH1, HtsH2, and HtsH3 can directly target the promoter of HSAF biosynthesis genes to regulate HSAF production in *L. enzymogenes*. The solid arrow indicates the demonstrated direct signal modulation. The dashed arrow suggests a potential signal regulation pathway. The dashed box represents the deletion of the *htsH1*, *htsH2* and *htsH3* genes”.

o It is unusual to have a membrane bound protein bind a transcription factor directly as proposed here. While the data seems reasonable it seems like it certainly deserves increased discussion

Response: Thank you for your kind comments. There are examples showing that membrane-bound proteins can directly bind to gene promoters. please refer to the manuscript entitled "ChiS is a noncanonical DNA-binding hybrid sensor kinase that directly regulates the chitin utilization program in *Vibrio cholerae*" (Klancher *et al.*, 2020).

o The studies seem disconnected from the role fo the system in actual QS. How this system responds in the presence of DSF is critical to understanding how this functions in soil.

Response: Thank you for your comments. We quantified HSAF production in the $\Delta rpfF$ and $\Delta rpfG$ mutant stains grown in 10% TSB medium or 10% TSB medium supplemented with 10 μ M canonical DSF. As shown in Figure S1, HSAF production in the $\Delta rpfF$ and $\Delta rpfG$ mutant strains was completely suppressed, and DSF significantly promoted HSAF production by the $\Delta rpfF$ mutant strain but did not promoted that by the $\Delta rpfG$ mutant strain. These findings suggested that DSF type-based QS systems are critical for regulating the biosynthesis of HSAF in *L. enzymogenes*. Please refer to lines 103-112 of the new manuscript.

o Similarly, how does RpfG association with these other proteins influence DSF perception? A better explanation of the QS circuit they are asserting this operates through needs to be presented.

Response: We thank the reviewer for the suggestion to help us improve this work. We revised the sentence as follows in lines 51-64 of the manuscript: “Several lines of evidence indicate that RpfC

and RpfG constitute a TCS responsible for the detection and transduction of the QS signal DSF. RpfC undergoes autophosphorylation upon sensing high levels of extracellular DSF signals. A previous study revealed that RpfG contains both an N-terminal response regulator domain and a C-terminal HD-GYP domain. The activated HD-GYP domain of RpfG has cyclic dimeric GMP (c-di-GMP) phosphodiesterase (PDE) activity that can degrade c-di-GMP, an inhibitory ligand of the global transcription factor Clp. Consequently, derepressed Clp drives the expression of several hundred genes, including those encoding virulence factor production in the plant pathogen *Xanthomonas*. The DSF signal family is a novel structural class of QS signals with the cis-2-unsaturated fatty acid moiety. Surprisingly, a DSF-like signal (LeDSF3), unlike other members of the DSF family, does not contain the cis double bond, and has been characterized as a QS signal in the biocontrol agent strain *Lysobacter enzymogenes*. However, the regulatory mechanism of the DSF-mediated QS system remains unknown in bacteria that are beneficial to plants”.

- Are there other experiments that would strengthen the paper further? How much would they improve it, and how difficult are they likely to be?

- o The exogenous addition of DSF to evaluate its effects on HSAF production or a reference to this work previously done elsewhere would do a better job of framing how this process works in the context of the role of QS normally held by RpfG. This should not be difficult.

Response: Thank you for your comments. We quantified HSAF production in the $\Delta rpfF$ and $\Delta rpfG$ mutant stains grown in 10% TSB medium or 10% TSB medium supplemented with 10 μ M canonical DSF. As shown in Figure S1, HSAF production in the $\Delta rpfF$ and $\Delta rpfG$ mutant strains was completely suppressed, and DSF significantly promoted HSAF production by the $\Delta rpfF$ mutant strain, but did not promoted that by the $\Delta rpfG$ mutant strain. These findings suggested that DSF type-based QS systems are critical for regulating the biosynthesis of HSAF in *L. enzymogenes*. Please refer to lines 103-112 of the new manuscript.

- o How does the presence of DSF influence the association between RpfG and HtsH1-3?

Response: Considering the reviewer’s suggestion, we used a pull-down assay to examine whether the presence of DSF influenced the association between RpfG and the HyTCS proteins (HtsH1, HtsH2 and HtsH3). The assay showed that the presence of 10 μ M DSF did not affect the interaction of RpfG with HtsH1, HtsH2 and HtsH3. One possible explanation is that DSF does not

directly participate in signal regulation downstream of RpfG.

• Are the claims appropriately discussed in the context of previous literature?

o This is the co-opting/reprogramming of an existing QS circuit to function in a decidedly non-QS dependent manner. This departure needs further discussion. There are lots of permutations to QS circuits such as LasR/QscR, TraM and other unusual regulators. This seems unique in being hijacked from the QS strategies. It really needs a better discussion in context of how their results impacts our understanding of QS.

Response: We thank the reviewer for the suggestion to help us improve this work. We have added the following statement in lines 270-288 of the manuscript: “Quorum sensing (QS) allows populations of bacteria to communicate via the exchange of chemical signals, resulting in coordinated gene expression in response to cell density. AHL signalling was first discovered in the marine bacterium *Vibrio fischeri*. In *V. fischeri*, LuxI and LuxR produce and respond to 3OC6-HSL, respectively. In addition to *V. fischeri*, *Pseudomonas aeruginosa* has emerged as an important model organism for QS research. In *P. aeruginosa*, the LasIR system produces and responds to 3OC12-HSL, and the RhIR system produces and responds to C4-HSL. QscR is an orphan LuxR receptor that is not linked to a luxI synthase gene. QscR responds to 3OC12-HSL produced by LasI. The quorum-sensing transcriptional activator TraR of *Agrobacterium tumefaciens*, which controls the replication and conjugal transfer of the tumour-inducing (Ti) virulence plasmid, is inhibited by the TraM antiactivator. In the DSF-mediated quorum sensing system, RpfC undergoes autophosphorylation upon sensing accumulated extracellular DSF signals. Through the conserved phosphorelay mechanism, RpfG is phosphorylated, which leads to activation of its c-di-GMP phosphodiesterase activity. Degradation of c-di-GMP releases Clp, which regulates subsets of virulence genes directly or through the downstream transcription factors FhrR and Zur. In this study, we revealed a new signal pathway by which the DSF type-based QS system component protein RpfG interacts with HtsH1, HtsH2 and HtsH3 to regulate the biosynthesis of HSAF in *L. enzymogenes*”.

• If the manuscript is unacceptable in its present form, does the study seem sufficiently promising that the authors should be encouraged to consider a resubmission in the future?

o While a better discussion, and possibly additional experiments, are required to address the connection to QS, they should be encouraged to resubmit.

Response: Thank you for your comments. We have carefully revised our paper. We hope that the revised manuscript meets with your approval.

References

Klancher, C.A., S. Yamamoto, T.N. Dalia & A.B. Dalia, (2020) ChiS is a noncanonical DNA-binding hybrid sensor kinase that directly regulates the chitin utilization program in *Vibrio cholerae*. *Proceedings of the National Academy of Sciences* 117: 20180-20189.

Reviewer #3 (Remarks to the Author):

This paper, for the first time, studied three hybrid two-component system (HyTCS) proteins, HtsH1, HtsH2, and HtsH3 in *Lysobacter*, and showed many interesting results. RpfG, the sensor protein with PDE activity in DSF mediated QS system, can interact directly with HtsH. The authors also provide convincing evidence that the phosphorylated HtsHs can interact with promoter of the HSAF synthetic cluster to regulate HSAF amount, while *htsHs* gene deletion cause no effects on growth.

Response: Thank you for your insightful comments and suggestions.

Specific comments:

1. The authors showed the evidence that all the three *htsHs* are in one transcript, indicating the only promoter located upstream of *htsH1*. The author also constructed a serials of *htsHs* mutants, then the expression of *htsH2* or *htsH3* may preserve or be affected in *htsH1* deletion mutant. The author should clarify that expression of *htsHs* downstream preserved.

Response: We thank the reviewer for the suggestion to help us improve this work. Using quantitative RT-PCR (qRT-PCR), we measured the mRNA abundance of *htsH1*, *htsH2*, *htsH3*, Le3074 and Le3075 in the *htsHs* mutants ($\Delta htsh1$, $\Delta htsh2$, $\Delta htsh3$ and $\Delta htsh123$) and found that the genes downstream of the *htsHs* gene deletion mutants were expressed (Figure S5). Please refer to lines 197-200 of the new manuscript.

2. All the three HtsHs are membrane-bounded proteins, as shown in Fig. 8, while the authors showed they bind to the promoter of HSAF synthetic gene cluster directly. Is there other examples that membrane-bounded proteins can interact with promoter? or other intermediate protein transfers the activation or inhibition effect?

Response: Thank you for your kind comments. There are examples showing that

membrane-bounded protein can directly bind to gene promoters. please refer to the manuscript entitled "ChiS is a noncanonical DNA-binding hybrid sensor kinase that directly regulates the chitin utilization program in *Vibrio cholerae*" (Klancher *et al.*, 2020).

3. The authors confirmed that RpfG binds directly to the HtsHs by pull-down assay, and HtsHs bind to the *lafB* promoter and positively regulate HSAF production. However, how RpfG affects HtsHs to bind the *lafB* promoter and positively regulate HSAF production?

Response: We thank the reviewer for the suggestion to help us improve this work. In this study, we found that RpfG interacts with the HtsH1, HtsH2 or HtsH3 protein and HtsH1, HtsH2, and HtsH3 can interact with each other. We also found that knockout of *rpfG* or *htsH1*, *htsH2*, and *htsH3* significantly reduced HSAF production. These results seem to confirm that RpfG forms a complex with HtsH1, HtsH2 and HtsH3 to regulate HSAF biosynthesis. However, the mechanism needs to be elucidated in the future. Please refer to lines 345-350 of the new manuscript.

4. HtsHs bind to the *lafB* promoter, but what DNA sequence in *lafB* promoter interact with HtsHs directly?

Response: Thank for your helpful comments. In this study, we performed an *E. coli*-based one-hybrid assay, electrophoretic mobility shift assay and surface plasmon resonance assay, and found that HtsH1, HtsH2, and HtsH3 could directly target the promoters of the HSAF biosynthesis gene. To further clarify the DNA sequence of the *lafB* promoter that interacts with HstHs, we performed DNaseI footprinting. However, we failed to obtain the DNA sequence of the *lafB* promoter that interacts with HtsHs directly, so the HstHs binding sequence still needs to be elucidated in the future.

5. The authors showed phosphorylated HtsH1, HtsH2, and HtsH3 positively regulate HSAF biosynthesis, could you show residues which amino acid residue in HtsHs was phosphorylated?

Response: Thank you for your insightful comments and suggestions. The biological functions of three consecutive HyTCS proteins have not been reported in bacteria. We report for the first time the biological functions of three consecutive HyTCS proteins. So we designated the HyTCS proteins name as HtsH (hybrid two-component signalling system regulating HSAF production)

based on the findings of this study. We are very sorry, we do not have enough information to predict the phosphorylated amino acid residues of HtsH1, HtsH2, and HtsH3.

Line 98: the word “of” should be deleted in “the production of in....”

Response: I apologize for this error. We have corrected it.

Line 145: “the cytoplasmic fragments of HtsH1, HtsH2, and HtsH3 (HtsH1C-Flag-His, HtsH2C-HA-His, and HtsH3C-Myc-His, respectively”, why used three different tag (flag,HA, Myc) fused into the three proteins ?

Response: Thank you for your comments. The three tags are respectively fused to the three proteins to better distinguish the HtsH1C, HtsH2C and HtsH3C proteins in the pull-down assay.

Line 152: “HtsH1, HtsH2, or HtsH3 proteins” should be clearly marked as HtsH1C....

Response: This was an error, and we have corrected it.

Line 199: Why choose the first gene *lafB* in the HSAF biosynthesis operon (*PlafB*)? *LafB* is the key enzyme for HSAF synthesis? Does HtsHs bind to other promoter in the operon?

Response: Thank you for your insightful comments and suggestions. The HSAF biosynthesis operon constitutes a single transcription unit (ORF1-ORF7) (Wang *et al.*, 2017). At the same time, there are many articles confirming that the promoter of the HSAF synthetic operon is *plafB* (the upstream region of ORF7) (Xu *et al.*, 2018, Xu *et al.*, 2021b, Xu *et al.*, 2021a). Therefore, to avoid unnecessary misunderstandings, we revised our statement as follows in lines 219-220 of manuscript: “As shown in Figure 6A, HtsH1, HtsH2, and HtsH3 directly bound to the upstream region of the HSAF biosynthesis operon (*plafB*)”.

Map of the HSAF gene cluster (Lou *et al.*, 2011)

References

- Klancher, C.A., S. Yamamoto, T.N. Dalia & A.B. Dalia, (2020) ChiS is a noncanonical DNA-binding hybrid sensor kinase that directly regulates the chitin utilization program in *Vibrio cholerae*. *Proceedings of the National Academy of Sciences* 117: 20180-20189.
- Lou, L., G. Qian, Y. Xie, J. Hang, H. Chen, K. Zaleta-Rivera, Y. Li, Y. Shen, P.H. Dussault, F. Liu & L. Du, (2011) Biosynthesis of HSAF, a tetramic acid-containing macrolactam from *Lysobacter enzymogenes*. *Journal of the American Chemical Society* 133: 643-645.

- Wang, P., H. Chen, G. Qian & F. Liu, (2017) LetR is a TetR family transcription factor from *Lysobacter* controlling antifungal antibiotic biosynthesis. *Applied microbiology and biotechnology* 101: 3273-3282.
- Xu, G., S. Han, C. Huo, K.H. Chin, S.H. Chou, M. Gomelsky, G. Qian & F. Liu, (2018) Signaling specificity in the c-di-GMP-dependent network regulating antibiotic synthesis in *Lysobacter*. *Nucleic acids research* 46: 9276-9288.
- Xu, K., D. Shen, S. Han, S.H. Chou & G. Qian, (2021a) A non-flagellated, predatory soil bacterium reprograms a chemosensory system to control antifungal antibiotic production via cyclic di-GMP signalling. *Environmental microbiology* 23: 878-892.
- Xu, K., D. Shen, N. Yang, S.H. Chou, M. Gomelsky & G. Qian, (2021b) Coordinated control of the type IV pili and c-di-GMP-dependent antifungal antibiotic production in *Lysobacter* by the response regulator PilR. *Molecular plant pathology* 22: 602-617.

REVIEWERS' COMMENTS:

Reviewer #1 (Remarks to the Author):

I confirmed that the authors had taken appropriate actions to address my points. These revisions will, as I hoped, help make the authors' arguments clearer and more accessible to readers. I look forward to seeing the final version of this paper when it is published.

Reviewer #3 (Remarks to the Author):

In the revised manuscript (ID: COMMSBIO-21-0536) the authors have added new data. The authors' data largely support the hypothesis that *Lysobacter* reprograms a DSF-mediated QS systems to regulate antifungal weapon (heat-stable antifungal factor, HSAF) production in a cyclic-di-GMP-independent manner. The additional data reinforced our understanding about bacterial QS systems. Overall, this is a promising manuscript with good quality of writing and straightforward logic. Therefore, I recommend acceptance of the article for publication in *Communications Biology*.

List of Responses

Dear Reviewers:

Thank you for your letter and for the reviewers' comments concerning our manuscript entitled "A predatory soil bacterium reprograms a quorum sensing signal system to regulate antifungal weapon production in a cyclic-di-GMP-independent manner" (ID: COMMSBIO-21-0536A). These comments are all valuable and very helpful for revising and improving our paper and provide important guidance in our research.

Responses to the reviewers' comments:

Reviewer #1 (Remarks to the Author):

I confirmed that the authors had taken appropriate actions to address my points. These revisions will, as I hoped, help make the authors' arguments clearer and more accessible to readers. I look forward to seeing the final version of this paper when it is published.

Response: We thank the reviewer for the nice comment.

Reviewer #3 (Remarks to the Author):

In the revised manuscript (ID: COMMSBIO-21-0536) the authors have added new data. The authors' data largely support the hypothesis that *Lysobacter* reprograms a DSF-mediated QS systems to regulate antifungal weapon (heat-stable antifungal factor, HSAF) production in a cyclic-di-GMP-independent manner. The additional data reinforced our understanding about bacterial QS systems. Overall, this is a promising manuscript with good quality of writing and straightforward logic. Therefore, I recommend acceptance of the article for publication in *Communications Biology*.

Response: We thank the reviewer for the nice comment on our work.